# Assessment of Respiratory Health Symptoms and Asthma in Children near a Drying Saline Lake

**DOI:** 10.3390/ijerph16203828

**Published:** 2019-10-11

**Authors:** Shohreh F. Farzan, Mitiasoa Razafy, Sandrah P. Eckel, Luis Olmedo, Esther Bejarano, Jill E. Johnston

**Affiliations:** 1Department of Preventive Medicine, Keck School of Medicine of University of Southern California, 2001 N. Soto Street, MC 9237, Los Angeles, CA 90089, USA; razafy@usc.edu (M.R.); eckel@usc.edu (S.P.E.); jillj@usc.edu (J.E.J.); 2Comite Civico del Valle, Brawley, CA 92227, USA; luis@ccvhealth.org (L.O.); esther@ccvhealth.org (E.B.)

**Keywords:** asthma, children’s respiratory health, rural areas, Imperial County, California, environment

## Abstract

Residents of the Imperial Valley, a rural, agricultural border region in California, have raised concerns over high rates of pediatric asthma symptoms. There is an urgent need to understand the influences and predictors of children’s respiratory health in Imperial Valley. We assessed the impacts of sociodemographic, lifestyle, and household factors on children’s respiratory health and asthma prevalence by administering a survey to parents of elementary school children (*n* = 357) in northern Imperial Valley. We observed an overall asthma prevalence of 22.4% and respiratory symptoms and allergies were widely reported, including wheezing (35.3%), allergies (36.1%), bronchitic symptoms (28.6%), and dry cough (33.3%). Asthmatics were significantly more likely to report respiratory symptoms, but high rates of wheezing, allergies, and dry cough were observed among nonasthmatics, suggesting the possibility for underdiagnosis of respiratory impairment in our school-age population. Having an asthmatic mother and exposure to environmental tobacco smoke were also associated with greater odds of asthma. Our findings provide evidence to support community concerns about children’s respiratory health, while also suggesting that household and demographic characteristics have limited explanatory power for assessing asthma in this population. This work provides critical baseline data with which to evaluate local environmental factors and their influence on asthma and respiratory symptoms.

## 1. Introduction

Over six million children in the United States (USA) are living with asthma, making it the most commonly diagnosed chronic childhood disorder [1]. In 2015, the reported prevalence of asthma among U.S. children was 8.4%, with rates ranging up to 13.4% among minority populations [1]. Although children of Mexican heritage have an overall lower prevalence of asthma compared to other groups of U.S. children, recent trends further suggest widening racial, ethnic, and economic disparities in terms of pediatric asthma prevalence, which may be in part related to environmental factors [2,3,4,5,6]. For example, the pediatric asthma prevalence observed among USA–Mexico border populations is frequently much higher than the national average of 8% for Hispanic children, although the reasons for these population differences remain largely unknown [7]. 

There is limited research on the prevalence of asthma and related risk factors in predominately Mexican-American rural communities in the USA. One such community in the rural agricultural valley situated between the USA–Mexico border and a drying saline lake in southeastern California (CA) has raised concern over elevated rates of pediatric asthma symptoms in their region. In this primarily low-income, majority Mexican-American region known as the Imperial Valley, the rate of asthma-related emergency room visits and hospitalizations for children ages 0–17 years is double the CA state average [8]. In 2005, a CA Health Department-led study in the region observed an asthma prevalence of 20% among middle school students and high rates of respiratory symptoms among nonasthmatics, suggesting that undiagnosed asthma was potentially common [8]. To date, the factors contributing to the high rates of adverse childhood respiratory health conditions in Imperial Valley have not been explored.

To begin to assess the factors influencing respiratory health in this unique population, we evaluated the current prevalence of asthma and other respiratory health conditions among elementary school children living in the northern Imperial Valley, CA. We further sought to determine whether asthma prevalence rates varied across sociodemographic and household characteristics. We focused on the northern end of the valley, in response to concerns regarding changing weather patterns, droughts, and competing water demands that are poised to cause the rapid shrinking of the Salton Sea, a 350-square-mile land-locked saline “sea” with the potential to significantly impact the future respiratory health and quality of life for nearby residents [9,10,11,12]. The recent enactment of a federally mandated water transfer agreement at the end of 2018 reduced the allotment of Colorado River water for the Imperial Valley, diverting water resources to nearby urban regions and effectively ending the supply of water to the Salton Sea [13]. This reallocation of water resources has resulted in an increasingly retreating shoreline leaving behind exposed playa, which has the potential to generate wind-blown dust, adding to ongoing community concerns about the already poor local air quality from contributors such as agricultural burning practices and cross-border particulate matter sources. The drying of the Salton Sea has unknown public health implications and the existing vulnerabilities of nearby populations are largely unmeasured. Characterizing the current respiratory health risk factors in this population could inform public health protection in the face of changing environmental factors. 

## 2. Methods

The Imperial Valley, located along the USA–Mexico border in Imperial County, CA, primarily consists of agricultural fields and open, undeveloped desert land, populated by a handful of small cities and towns [14,15]. Imperial County is considered a rural medical service study area (MSSA), defined as an area with fewer than 250 persons per square mile and no population center exceeding 50,000, and parts of northern Imperial County are defined as a frontier MSSA, indicating a rural MSSA with fewer than 11 residents per square mile [16,17]. A cross-sectional school-based study was conducted between May 2017 and May 2018 in Imperial County, CA. Four elementary schools across four towns within approximately 20 miles of the Salton Sea were invited to participate. In three of the four towns, a single elementary school services the entire town; in the fourth town, a representative elementary school was selected with input from community partners. Study staff visited classrooms to introduce the study to the children and distribute the study packets. All surveys, materials, and consent forms were provided in both English and Spanish, and children returned all study materials to the classroom within one week. Parents and guardians completed a 64-item health survey, adapted from validated questionnaires from the International Study of Asthma and Allergies in Childhood (ISAAC) [18], a questionnaire developed to assess asthma prevalence that has been validated in populations around the world, as well as questions used as part of the Southern California Children’s Health Study [19]. 

Demographic and lifestyle questions assessed the children’s characteristics such as age, sex, race/ethnicity, as well as parent/caregiver information including annual household income, educational attainment, insurance coverage, and tobacco usage. Questions on the child’s medical and health history focused on the occurrence of severe respiratory illnesses before the age of two, asthma diagnoses, and recent respiratory conditions and symptoms. Respiratory medication use was assessed using photographic charts of common asthma rescue and controller medications. Questions about the home environment included the presence of pets or farm animals, as well as indoor housing conditions (presence of mold, carpeting, water damage, pests, home heating and cooling, humidifier use, and gas stove use).

Health surveys were distributed to 429 children during two week-long distribution periods over the data collection year between May 2017 and May 2018, with an overall participation rate of 83.2% (357 of 429). For the purposes of this study, if a survey was filled out for a child during both rounds of data collection, the first survey was used in these analyses. Written informed consent was obtained from a parent or guardian. All protocols, consent forms, and survey materials were approved by the University of Southern California Institutional Review Board (HS-17-00204).

### Statistical Analysis 

All analyses were conducted using STATA IC Version 15 (Stata, College Station, TX, USA). Chi-square tests were used to test for associations between categorical variables. Logistic regression was used to examine associations between potential risk factors and binary outcome variables (physician-diagnosed asthma or the presence of respiratory symptoms). We examined the associations in unadjusted models and stepwise adjustment for a priori selected covariates (minimally adjusted model: age and sex, language of survey response) and fully-adjusted final models (adjusted for age, sex, language of survey response, education level of the parent/caregiver survey respondent, school of enrollment). Statistical significance was defined as *p* < 0.05.

Children were classified as asthmatic if the child’s parent answered affirmatively to the question “Has your child ever received an official doctor’s diagnosis for asthma?” A child was considered to have bronchitic symptoms if the child’s parent reported at least one of the following: 1) daily cough for three months in a row, 2) congestion or phlegm for at least three months in a row, or 3) bronchitis in the past year [20]. Health insurance was categorized into public, private, or none. Respiratory medications were categorized as rescue, control, or other [21].

## 3. Results

A total of 357 students participated in this study (Table 1). The mean age was 7.6 (SD: 0.86) years. Our sample included slightly more female (54.9%) than male children (45.1%). Most parents completed the questionnaires in English (64.2%), with almost all participants self-identifying as Latino/a (93.3%). Most reported having health insurance (90.5%), with the majority enrolled in a public health insurance plan (70.3%). Approximately one-third of participants reported a total household of less than $15,000 a year (30.0%), with a minority of participants earning above $50,000 (14.3%). Approximately 20% of the study population did not know or chose not to report their annual total household income. Of the parents or caregivers who responded to the questionnaire, approximately 68% were the child’s biological mother, 19% were the child’s biological father, and the remaining 13% were other relatives or caregivers. About half of the parent or caregiver respondents had a high school education or less (50.4%). 

Among the participants, the overall prevalence of doctor-diagnosed asthma was 22.4%; 48.8% were boys and 51.3% were girls. Statistically significant differences between asthmatic and nonasthmatic children were observed for key demographic variables, including language of the survey, health insurance, and total household income (Table 1). We also observed significant differences between asthmatic and nonasthmatic children for respiratory health history variables including biological mother with asthma (25.1% versus 8.7%, *p* < 0.001), upper respiratory tract infections before age two (16.3% versus 2.9%, *p* < 0.001), and lower respiratory tract infections before age two (32.5% versus 7.9%, *p* < 0.001) (Table 2). Most doctor diagnoses of asthma had occurred when the child was between two and four years old (42.5%).

Respiratory symptoms and allergies were widely reported among this study population (Figure 1; Appendix A). The most frequently reported respiratory symptoms and related conditions in children were wheezing (35.3%), allergies (36.1%), bronchitic symptoms (28.6%), and a dry cough (33.3%) (Appendix A). Asthmatics were significantly more likely to report any respiratory symptoms, but among nonasthmatics we still observed high rates of wheezing (19.9%), allergies (27.4%), and dry cough (23.1%) (Figure 1). Among those who reported ever having wheezed, 7.9% stated that wheezing impacted their ability to speak in the past 12 months, 37.3% their ability to exercise in the past 12 months, and 17.5% their ability to sleep in the past 12 months, as defined as having woken up due to wheezing more than one night per week (Appendix A). 

We observed statistically significant differences between asthmatics and nonasthmatics in the use of healthcare resources and respiratory medications, including doctor visits due to wheezing, emergency room visits due to wheezing, rescue medication use, control medication, and nebulizer use (Figure 2). Among those who identified as asthmatic, 45.0% and 77.5% reported having gone to the emergency room or to a doctor, respectively, at least once in their lives for wheezing (Figure 2). The majority of asthmatics (82.5%) reported that they were currently taking asthma medication, with 80.0% taking rescue medication, 37.5% taking control medication, and 70.0% having used a nebulizer at least once in the past year. Among the nonasthmatic population, 14.8% of students had been to a doctor due to wheezing, 7.6% had to go to the emergency room for wheezing, and 9.4% are currently taking rescue asthma medication raising concerns for potential undiagnosed cases (Figure 2). 

Using logistic regression models, either minimally adjusted (age and sex) or fully adjusted for covariates (age and sex, plus language of survey, parental education, and school of enrollment), we examined the association between demographic and household risk factors with physician-diagnosed asthma (Table 3). Having a biological mother with asthma was associated with a nearly 3-fold greater odds of asthma (OR: 2.92; 95 CI: 1.44–5.93) in fully-adjusted models. Exposure to household environmental tobacco smoke was also associated with 4-fold greater odds of asthma (OR: 4.00; 95% CI: 1.21–13.23) in fully adjusted models. Asthma was positively associated with being enrolled in private insurance (OR: 2.05; 95% CI: 1.15–3.65) in minimally adjusted models, but this association was attenuated in the fully adjusted models and no longer statistically significant. 

We also investigated the influence of various housing characteristics on the asthma prevalence in our population (Table 3). Although most housing characteristics did not significantly differ between asthmatics and nonasthmatics (Appendix A), we did observe differences in asthma prevalence across minimally and fully adjusted models associated with length of cooking gas use in the home and the presence of pets. Asthma prevalence was marginally associated with using a gas stove for more than 1 h per day (OR: 2.58; 95% CI: 0.97–6.82) (Table 3) in fully adjusted models. About one-third of participants reported having a furry pet at home (29.4%) (Appendix A), which was observed to be negatively associated with odds of asthma (OR: 0.42; 95% CI: 0.21–0.82) in fully adjusted models (Table 3). 

## 4. Discussion

Our study sought to characterize the prevalence rates of asthma and other respiratory conditions among rural, predominantly Mexican-American elementary school children living near the Salton Sea in the Imperial Valley, CA prior to the implementation of the water transfer agreement poised to dramatically accelerate the shrinking of this saline lake. Using parent-reported survey information, we observed an overall asthma prevalence of 22.4%, which is significantly higher than the national average for children ages 5–11 years (8.8%), the national average for Mexican-American children ages 0 to 17 (6.2%), and the California state rates for children ages 0 to 17 (14.5%) [1,22,23]. In addition to asthma, we observed high rates of various respiratory symptoms, including wheezing (35%), allergies (36%), bronchitic symptoms (28%), and persistent dry cough (33%), which were widely reported among both asthmatic and nonasthmatic children. We also identified sociodemographic and household factors that may be related to asthma. The findings of this study provide evidence to support community concerns about children’s respiratory health, suggesting that household and demographic characteristics have limited explanatory power for assessing asthma in this population. Our study provides critical baseline data with which to evaluate the effect of local environmental exposures on asthma and respiratory symptoms.

A small number of similar studies have investigated asthma prevalence and respiratory health in the USA–Mexico border areas. Together, these studies collectively suggest higher rates of asthma and respiratory illness along the U.S. side [6,24,25,26]. Our findings were similar to those of several other studies among children living in largely urban border areas, including a study conducted on the Arizona–Sonora border, which estimated a 25.8% rate of asthma among adolescents aged 13–14 [6]. A study looking at the prevalence of children’s respiratory health conditions in El Paso, TX reported a similar asthma rate of 17% among fourth and fifth graders [24]. Our estimates are, however, much higher than those observed in other studies from Texas [25], and among similarly aged fifth-grade students in the Nogales, Mexico/Arizona border region [26], which found asthma prevalence rates ranging from 5.8% to 9.4%. To our knowledge, the Border Asthma and Allergies (BASTA) study is the only other study of respiratory health outcomes to have been conducted in the Imperial Valley; it found an asthma prevalence of 20% among students aged 13–14, which is comparable to our observed rate of 22.4%. However, compared to our study, the BASTA study found higher reported rates of allergies among participants (57% versus 36%) [8]. Among children living in the Coachella Valley, a rural community north of the Salton Sea, 17.5% of sampled children were asthmatic or had a persistent cough [27]. Although similarities exist across these various studies, caution must be exercised as other studies may have used different assessment instruments or criteria to determine asthma prevalence.

Our results also suggest the possibility of underdiagnosis of asthma and respiratory impairment in our school-age population. A number of studies have suggested that there may be a substantial number of children who report asthma-like symptoms but remain undiagnosed [28,29,30,31,32]. Prior work in USA–Mexico border communities found that, among those who reported wheezing, there was a high risk of possible undiagnosed asthma [22,31]. In our sample of school children, nearly one in five nonasthmatics reported having at least one episode of wheezing in their lifetime, nearly one in seven reported having ever visited the doctor due to wheezing, and about one in 13 reported ever having gone to the ER due to wheezing. Substantial proportions of nonasthmatic children reported a persistent dry cough at night, bronchitic symptoms, or the recent use of an asthma rescue medication. Among nonasthmatics who reported having at least one episode of wheezing in their lifetime, half experienced wheezing in the past 12 months, and a quarter experienced wheezing after exercise. Given the large number of children in our population who reported respiratory symptoms without an asthma diagnosis, this suggests more widespread respiratory impairment in this population.

Prior research found that children not diagnosed with asthma, but reporting wheezing-like symptoms, experienced missed school days, limitation of physical activities, and sleep disturbances [29]. Similar to our findings, 7% of children with current asthma-like symptoms but no diagnosis in a North Carolina-based school study reported an emergency department visit due to an episode of wheezing [29]. In a similar study conducted by van Gent et al., children with undiagnosed asthma had lower quality-of-life scores than healthy control individuals, and also experienced, on average, a more than one week longer absence from school over 12 months because of respiratory symptoms compared to healthy individuals [28]. Children may have increased vulnerability to environmental insults, particularly in this region, and future work will further investigate the impact of breathing difficulties on quality of life across both asthmatic and nonasthmatic children in our study population [33].

Our health survey collected information from parents on a number of sociodemographic, lifestyle, and household characteristics that may be related to children’s respiratory health. When we examined maternal health factors, we found that having a biological mother with asthma was strongly related to child asthma and was consistent across all models tested. This is expected as family history and genetic predisposition are important components of asthma risk [34,35]. Additionally, as reported by numerous studies, we observed that maternal smoking during pregnancy increased the odds of child asthma; however, this was based on a relatively small number of mothers who reported smoking, and this association did not reach statistical significance [36,37,38].

Conditions in the indoor home environment have frequently been associated with children’s asthma risk and respiratory symptoms. In line with previous studies, we observed a strong positive association between exposure to household environmental tobacco smoke and asthma diagnosis [34,39,40]. We also observed a positive association between the use of a gas stove for more than 1 h per day and asthma diagnosis. Previous studies have found that gas stoves release respiratory irritants and may increase the risk of respiratory symptoms and asthma exacerbations in children [41]. Although other studies have reported other significant risk factors in the home such as water damage or carpeting, we did not observe significant associations between these variables and asthma in our analyses [19,27,34,35]. Having a furry pet at home was negatively associated with asthma in our study population. This is contrary to some other findings, although pet ownership has been inconsistently associated with asthma and allergic symptoms in various studies [42,43,44]. Despite this being a heavily agricultural community, no significant associations were found between reporting an asthma diagnosis and having regular contact with a farm animal. Further investigation of the role of exposure to pets and other animals in asthma is warranted, given the conflicting findings across studies.

Our study of rural, primarily Mexican-American elementary school children provides novel insights into the factors impacting respiratory health in the Imperial Valley. Our assessment benefited from the collection of numerous demographic, lifestyle, and home characteristics and the use of validated questionnaires to assess a variety of respiratory health symptoms. However, our study was also limited by several factors. In this study, we relied on the respondents’ questionnaire answers to assess asthma diagnosis, rather than medical records or physical examination. It is possible that defining asthma according to self-reported doctor diagnosis may have resulted in some outcome misclassification among our participants. Furthermore, it is possible that parents/guardians of asthmatics or those who were concerned about their child’s respiratory health were more likely to complete the survey than parents of nonasthmatic children, which could have led to over- or under-representation of symptoms and exposures. Lastly, although we examined numerous potential factors, we cannot rule out the possibility of residual confounding in our models. The limited ability of demographic, lifestyle, and other known risk factors, such as housing characteristics, to explain the high observed rates of asthma (22.4%) and respiratory symptoms in this young population suggests that environmental factors are likely contributors.

Rural communities in the southwestern USA and along the USA–Mexico border face changing weather patterns, droughts, and competing water demands that are dramatically altering the landscape and creating conditions conducive to the production of wind-blown dust. As the Salton Sea recedes, primarily due to local changes in water supplies and reductions in agricultural runoff, it is anticipated that 40–80 more tons of dust per day will be released into the local environment [45], exposing large swaths of playa and generating wind-blown dust [13]. Because the particulates from dried lakebeds have been shown to be smaller in size, they are more easily respirable by humans [46], and such substantial increases in dust have the potential to significantly impact the respiratory health and quality of life for nearby residents [9,10,11,12]. This impact is expected to be seen first along the southern edge of the Sea, where our findings show that, overall, children are already disproportionally affected by asthma. Follow-up studies of these children are underway to explore the effects of a changing environment on respiratory health.

Our findings may help provide a better understanding of the health needs of the community, as well as strategies to inform community-specific preventive programs to address the symptoms of asthma and respiratory impairment. The disappearance of the Salton Sea will likely have unforeseen public health implications, while children and people with preexisting health conditions, such as asthma, may be more vulnerable to the impacts of such environmental changes [13,47,48]. Future research will longitudinally examine children’s respiratory health, incorporating both ambient exposure measurements and physiological assessments over time. While this work began as a way to provide a baseline assessment of children’s health in relation to the shrinking of Salton Sea, the broad implications of the present research reveal a public health and environmental justice crisis that requires action, attention, and further research to protect the health and wellbeing of local communities. 

## Figures and Tables

**Figure 1 ijerph-16-03828-f001:**
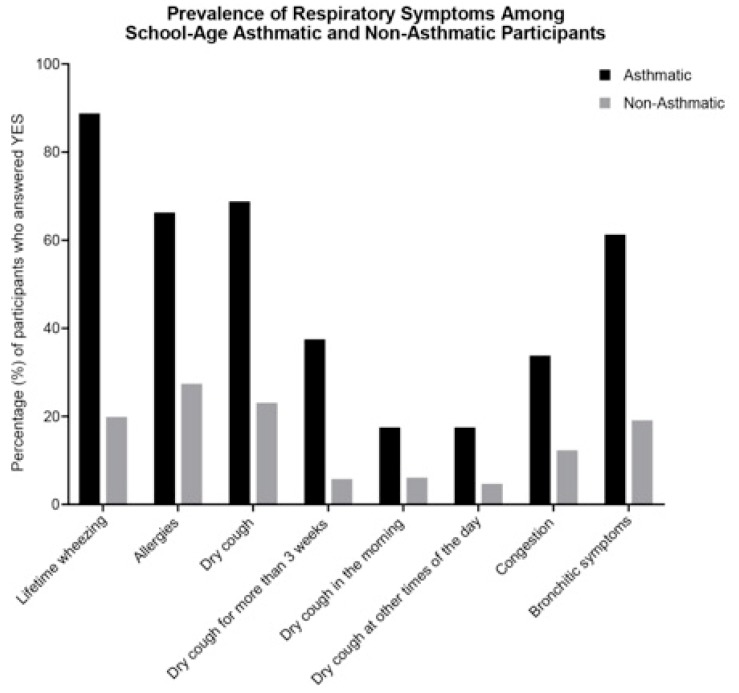
Prevalence of reported respiratory symptoms among school-age participants. Bars represent percentage of asthmatics (*n* = 80; black bars) and nonasthmatics (*n* = 277; gray bars) who were reported to have any of the listed respiratory symptoms in the past 12 months, unless otherwise specified.

**Figure 2 ijerph-16-03828-f002:**
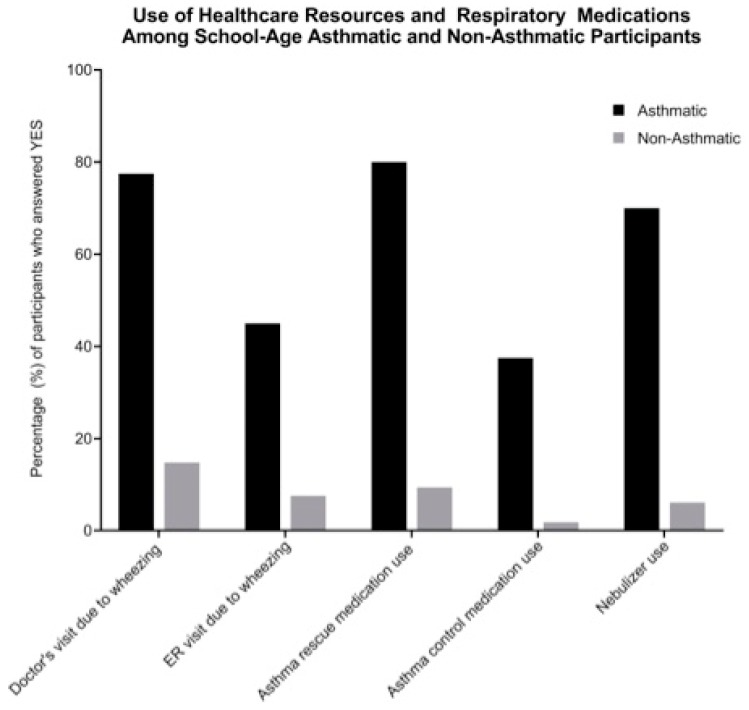
Use of healthcare resources and respiratory medications among school-age asthmatic and nonasthmatic participants. Bars represent percentage of asthmatics (*n* = 80; black bars) and nonasthmatics (*n* = 277; gray bars) who were reported to have utilized any of the listed healthcare resources or medications in the past 12 months.

**Table 1 ijerph-16-03828-t001:** Demographic characteristics of asthmatic and nonasthmatic study participants (*N* = 357).

Demographic Characteristics	All *N* = 357 (%)	Asthmatics *N* = 80 (%)	Nonasthmatics *N* = 277 (%)	*p*-Value ^a^
**Language of survey**				
English	229 (64.2)	59 (73.8)	170 (61.4)	0.04
Spanish	128 (35.8)	21 (26.3)	107 (38.6)	
**Sex**				
Female	196 (54.9)	41 (51.3)	155 (56.0)	0.46
Male	161 (45.1)	39 (48.8)	122 (44.0)	
**Race/Ethnicity**				
Latino	333 (93.3)	74 (92.5)	259 (93.5)	0.38
White	18 (5.0)	5 (6.3)	13 (4.7)	
Black	3 (0.8)	0 (0.0)	3 (1.1)	
Native American	1 (0.3)	1 (1.3)	0 (0.0)	
Other	2 (0.6)	0 (0.0)	2 (0.7)	
**Age**				
5–6 years	32 (9.0)	9 (11.3)	23 (8.3)	0.69
7 years	131 (36.7)	28 (35.0)	103 (37.2)	
8 years	148 (41.5)	35 (43.8)	113 (40.8)	
9–10 years	46 (12.9)	8 (10.0)	38 (13.7)	
**Health Insurance**				
Public	251 (70.3)	51 (63.8)	200 (72.2)	0.01
Private	72 (20.2)	25 (31.3)	47 (17.0)	
None	34 (9.5)	4 (5.0)	30 (10.8)	
**Total Household Income**				
Less than $15,000	107 (30.0)	22 (27.5)	85 (30.7)	0.01
$15,000 to $29,999	71 (19.9)	15 (18.8)	56 (20.2)	
$30,000 to $49,999	54 (15.1)	8 (10.0)	46 (16.6)	
More than $50,000	51 (14.3)	21 (26.3)	30 (10.8)	
Don’t know	74 (20.7)	14 (17.5)	60 (21.7)	
**Parent/caregiver education ^b^**				
Less than 12th grade	90 (25.2)	13 (16.3)	77 (27.8)	0.08
Completed 12th grade	90 (25.2)	23 (28.8)	67 (24.2)	
Some college or technical school	122 (34.2)	30 (37.5)	92 (33.2)	
4 years of college or more	35 (9.8)	12 (15.0)	23 (8.3)	
Missing	20 (5.6)	2 (2.5)	18 (6.5)	
**Two-adult household**				
Yes	202 (56.6)	46 (57.5)	156 (56.3)	0.50
No	142 (39.8)	34 (42.5)	108 (39.0)	
Missing	13 (3.6)	0 (0.0)	13 (4.7)	

^a^ As determined by chi-squared or Fisher’s exact test. ^b^ Education level of survey respondent.

**Table 2 ijerph-16-03828-t002:** Respiratory health history of school-age asthmatic and nonasthmatic participants.

Health Characteristics	All *N* = 357 (%)	Asthmatics *N* = 80 (%)	Nonasthmatics *N* = 277 (%)	*p*-Value ^a^
**Biological mother has asthma**				
Yes	44 (12.3)	20 (25.1)	24 (8.7)	<0.001
No	304 (85.2)	57 (71.3)	247 (89.2)	
Missing	9 (2.5)	3 (3.8)	6 (2.2)	
**Biological mother smoked while pregnant with child**				
Yes	16 (4.5)	5 (6.3)	11 (4.0)	0.36
No	333 (93.3)	72 (90.0)	261 (94.2)	
Missing	8 (2.2)	3 (3.8)	5 (1.8)	
**Lower respiratory infection during infancy**				
Yes	48 (13.5)	26 (32.5)	22 (7.9)	<0.001
No	309 (86.6)	54 (67.5)	255 (92.1)	
**Upper respiratory infection during infancy**				
Yes	21 (5.9)	13 (16.3)	8 (2.9)	<0.001
No	336 (94.1)	67 (83.8)	269 (97.1)	
**Asthma during infancy**				
Yes	51 (14.3)	47 (58.8)	4 (1.4)	<0.001
No	306 (85.7)	33 (41.3)	273 (98.6)	
**Age of asthma diagnosis ^b^**				
Under 2 years old	--	28 (35.0)	--	--
2–4 years old	--	34 (42.5)	--	--
5 years old and above	--	18 (22.5)	--	--

^a^ As determined by chi-squared or Fisher’s exact test. ^b^ Among individuals reporting doctor diagnosis of asthma only.

**Table 3 ijerph-16-03828-t003:** Associations between individual demographic and household risk factors and the prevalence of asthma.

Characteristics	Minimally Adjusted ^a^ OR (95% CI)	*p*-Value	Fully Adjusted ^b^ OR (95% CI)	*p*-Value
Participant demographics				
Household Income ^c^	1.05 (0.89–1.24)	0.53	1.05 (0.88–1.25)	0.62
Health insurance				
**Public**	Ref.	--	Ref.	--
**Private**	2.05 (1.15–3.65)	0.02	1.68 (0.89–3.15)	0.11
**None**	0.53 (0.18–1.59)	0.26	0.49 (0.16–1.55)	0.23
Household environmental smoke				
**No**	Ref.	--	Ref.	--
**Yes**	4.24 (1.37–13.07)	0.01	4.00 (1.21–13.23)	0.02
Biological mother smoked while pregnant				
**No**	Ref.	--	Ref.	--
**Yes**	1.66 (0.55–4.94)	0.37	2.23 (0.59–8.30)	0.23
Biological mother has asthma				
**No**	Ref.	--	Ref.	--
**Yes**	3.79 (1.94–7.39)	<0.001	2.92 (1.44–5.93)	0.003
Play sports at least twice a week				
**No**	Ref.	--	Ref.	--
**Yes**	1.43 (0.85–2.42)	0.17	1.62 (0.93–2.83)	0.09
Housing characteristics				
Housing type				
**Home**	Ref.	--	Ref.	--
**Apartment**	0.97 (0.56–1.68)	0.97	1.00 (0.55–1.80)	0.99
**Mobile home or trailer**	0.86 (0.35–2.10)	0.74	1.73 (0.62–4.82)	0.29
Gas cooking stove				
**No**	Ref.	--	Ref.	--
**Yes**	0.59 (0.31–1.12)	0.11	0.65 (0.32–1.34)	0.25
Length of daily gas stove use				
**Less than 30 min**	Ref.	--	Ref.	--
**Less than 1 h**	1.45 (0.77–2.70)	0.25	1.42 (0.74–2.75)	0.29
**More than 1 h**	3.17 (1.31–7.65)	**0.01**	2.58 (0.97–6.82)	0.06
Home heater				
**No**	Ref.	--	Ref.	--
**Yes**	2.46 (1.06–5.68)	**0.04**	2.08 (0.87–4.94)	0.10
Water damage in home				
**No**	Ref.	--	Ref.	--
**Yes**	2.23 (1.00–4.98)	**0.05**	1.73 (0.74–4.02)	0.21
Mold in home				
**No**	Ref.	--	Ref.	--
**Yes**	1.37 (0.69–2.73)	0.37	1.21 (0.59–2.48)	0.60
Musty odor in home				
**No**	Ref.	--	Ref.	--
**Yes**	2.56 (0.85–7.74)	0.10	2.14 (0.67–6.88)	0.20
Carpet in home				
**No**	Ref.	--	Ref.	--
**Yes**	1.32 (0.78–2.25)	0.30	1.53 (0.86–2.72)	0.15
Child has lived in same house for whole life				
**No**	Ref.	--	Ref.	--
**Yes**	1.32 (0.77–2.30)	0.31	1.29 (0.71–2.35)	0.41
Pet ownership				
**None**	Ref.	--	Ref.	--
**Furry pet**	0.48 (0.26–0.91)	**0.02**	0.42 (0.21–0.82)	**0.01**
**Other pets**	1.17 (0.48–2.83)	0.74	0.87 (0.34–2.24)	0.77
Regular contact with farm animals				
**No**	Ref.	--	Ref.	--
**Yes**	1.21 (0.57–2.57)	0.62	1.52 (0.67–3.45)	0.32
Problem with rodents in the home				
**No**	Ref.	--	Ref.	--
**Yes**	0.90 (0.36–2.20)	0.81	0.84 (0.33–2.13)	0.72
Problem with insects in the home				
**No**	Ref.	--	Ref.	--
**Yes**	1.19 (0.66–2.16)	0.57	0.97 (0.51–1.82)	0.92

^a^ Minimally adjusted model includes age and sex as covariates. ^b^ Fully adjusted model includes age, sex, language, parental education, and school as covariates. ^c^ Modeled as a continuous variable.

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
