# Peer review of "Assessment of Respiratory Health Symptoms and Asthma in Children near a Drying Saline Lake"

_ijerph, 2019, doi:10.3390/ijerph16203828_

Round 1
Reviewer 1 Report
The authors present important information about the overall asthma prevalence and respiratory symptoms and allergies in this environmental justice community near the US-Mexico border. This paper provides important baseline data for this vulnerable population located near the Salton Sea. These populations are understudied.
I have the following specific suggestions to improve the manuscript.
It is important to include the English and Spanish versions of the questionnaire used in the supplemental section, so the reader can ascertain exactly how questions were worded. I didn’t see the univariate data on ‘Biological mother smoked while pregnant’ although it appears in the model results in Table 3. Please add this to supplemental material. Also, as data is extremely sparse in this population, please give breakdown of ‘Biological mother smoked while pregnant’ and environmental tobacco exposure in the home by ethnicity and income in supplemental materials. Prevalence of these factors is usually lower in Latinos. Did the authors ask about electronic cigarette use in the home? If not, this question should be added to future studies. Authors also should consider asking about tobacco and e-cig exposure outside the home. Does ‘parental education’ in Table 1 refer to the person answering the question? (the father vs mother?) If another person filled out the questionnaire, was this answer for the mother? Please explain. Why were parental education, school of enrollment used in the final model? Schools might have different exposure to pollution. Education might affect doctor diagnosis/ access to care and should be examined. The authors should provide a bit more context on environmental pollution for the community. Please add a few more sentences about the Salton Sea and why it is drying up, and environmental justice concerns. In addition, the reader might assume that the main exposure to the children is to Salton Sea dust. Please add a description of agricultural burning practices and other community exposures. In supplemental materials, please include the PM2.5 levels from nearby monitors for the 3 months prior to these surveys. Efforts are underway to reduce burning activities in the Imperial Valley, as well as burning activities in Mexico (open trash burning, agricultural burning). The authors state in Methods that “Health surveys were distributed to 429 children during two week-long distribution periods over the data collection year”. As air pollution (PM2.5 and PM10) varies greatly over the year, and in addition being inside air conditioning can reduce particulate exposure during the summer, please provide dates of surveys. Also present any evidence (for children for which you have two surveys) that symptoms about which children were asked regarding the prior three months did not vary by survey date. Also, did any children for which you have more than one survey change their address over the two surveys? Did symptoms change?
Important work overall – this data is very valuable to protect public health in these vulnerable communities.
Author Response
Reviewer 1
Please see our responses below each point in italics.
It is important to include the English and Spanish versions of the questionnaire used in the supplemental section, so the reader can ascertain exactly how questions were worded.All survey questions that were included here were adapted from the English and Spanish versions of the previously published questionnaires from the International Study of Asthma and Allergies in Childhood (ISAAC) and the Southern California Children’s Health Study, as referenced in our methods section (please see lines 77-79).
I didn’t see the univariate data on ‘Biological mother smoked while pregnant’ although it appears in the model results in Table 3. Please add this to supplemental material.
It appears that there was an issue in the display of Table 2 in the formatted manuscript version provided to reviewers and we apologize for this error. This information on ‘Biological mother smoked while pregnant’ is now included here in Table 2.
Also, as data is extremely sparse in this population, please give breakdown of ‘Biological mother smoked while pregnant’ and environmental tobacco exposure in the home by ethnicity and income in supplemental materials. Prevalence of these factors is usually lower in Latinos.
The prevalence of smoking in our study is very low with only 16 (4.5%) parents/caregivers reporting that the biological mother smoked during pregnancy and 13 (3.6%) reporting any current smoking in the child’s household. The prevalence of smoking by income and among mothers during pregnancy by race/ethnicity were very similar to our overall population, with 92.3% of Latinos and 7.7% of whites reporting household smoking and 93.8% of Latinos and 6.3% of whites reporting smoking mothers during pregnancy. Those who reported smoking in the home or among biological mothers were primarily those who reported an income of less than $30k (61% and 68%, respectively). As the overall prevalence of smoking during pregnancy and current smoking in the household was very low in this population, we have chosen not to include this breakdown by income and race/ethnicity in the results, but will add upon the editor’s request.
Did the authors ask about electronic cigarette use in the home? If not, this question should be added to future studies. Authors also should consider asking about tobacco and e-cig exposure outside the home.
The reviewer makes an important suggestion and although we did not ask about e-cigarette use in the home, nor did we ask about parental use of tobacco or e-cigarette products outside of the home in this survey, we will consider adding these questions to future surveys.
Does ‘parental education’ in Table 1 refer to the person answering the question? (the father vs mother?) If another person filled out the questionnaire, was this answer for the mother? Please explain.
We thank the reviewer for pointing out this ambiguity in the text. We have now clarified that the education level is that of the survey respondent (either a parent or caregiver), both in the table footnotes, as well as in the methods. As we note in the first paragraph of the results, the majority (68%) of survey respondents were mothers of the children, 19% were fathers and the remaining 13% were a primary caregiver.
Why were parental education, school of enrollment used in the final model? Schools might have different exposure to pollution.
In order to adjust for socioeconomic status, we included parental education as a covariate. Other SES variables, including health insurance and income were highly correlated with parental education, but parental education was selected as a representative variable to avoid over-adjustment due to our small sample size and because it had the strongest predictive value in our models. We also included school of enrollment, to account for variability across enrollment sites. It is possible that school site may be related to environmental exposures and this will be explored in forthcoming analyses.
Education might affect doctor diagnosis/ access to care and should be examined.
We thank the reviewer for this suggestion and although we did not collect information on access to care, the large majority of our population is insured (90.5%). We plan to explore the impacts of access to care on respiratory health in future surveys.
The authors should provide a bit more context on environmental pollution for the community. Please add a few more sentences about the Salton Sea and why it is drying up, and environmental justice concerns.
We thank the reviewer for this suggestion and have now added additional description of the context for this imminent issue to the Introduction (see lines 101-108).
In addition, the reader might assume that the main exposure to the children is to Salton Sea dust. Please add a description of agricultural burning practices and other community exposures.
We have taken the reviewer’s important suggestion and have now added description of additional contributors to air pollution concerns to the Introduction (see lines 105-108).
In supplemental materials, please include the PM2.5 levels from nearby monitors for the 3 months prior to these surveys. Efforts are underway to reduce burning activities in the Imperial Valley, as well as burning activities in Mexico (open trash burning, agricultural burning).
We thank the reviewer for bringing up this important point. Unfortunately, little to no regulatory monitoring data for PM2.5 is available for our study area in northern Imperial Valley for the time period during which we conducted this study. Future work will incorporate real-time community air monitoring data and we have included a line in the discussion to indicate that this work is ongoing and follow-up of the children may help elucidate the role of environmental factors in respiratory health in this area.
The authors state in Methods that “Health surveys were distributed to 429 children during two week-long distribution periods over the data collection year”. As air pollution (PM2.5 and PM10) varies greatly over the year, and in addition being inside air conditioning can reduce particulate exposure during the summer, please provide dates of surveys.
We have now included additional details in the methods about the time frame of survey administration, which occurred at approximately the same time period and season in first in May 2017 and again in May 2018.
Also present any evidence (for children for which you have two surveys) that symptoms about which children were asked regarding the prior three months did not vary by survey date.
Our survey only asked about children’s symptoms in the past 12 months, so we do not have the ability to look at symptoms over a shorter period of time (i.e. 3 months) for consistency. As the number of children who returned two surveys is quite small and since surveys of the past 12 months of symptoms were administered approximately one year apart would be expected to vary year to year, we have chosen not to include this analysis in this manuscript.
Also, did any children for which you have more than one survey change their address over the two surveys? Did symptoms change?
We received very few surveys from the same children two years in a row and of those who did respond twice, very few children moved outside of their town within the survey year who were not lost to follow-up and remained at the same school. Therefore, we don't have adequate statistical power to examine changes in symptoms in relation to change of address at this time. Further, we did not assess time varying predictors of respiratory symptoms in this survey, but hope to incorporate these factors into future work to explore spatial variability in symptoms.
Reviewer 2 Report
The Imperial Valley is a rural region in California that has raised concerns of a high rate of pediatric asthma symptoms. In this cross-sectional study, Farzan et al. analyzed the respiratory health and asthma prevalence in a cohort of elementary school children (n=357), enrolled in schools located near the Salton Sea, in the Imperial Valley area. The authors found an overall asthma prevalence of 22.4% in the study population, which stated to be significantly higher than the national and state rates (8.88% and 14.5%, respectively) for children aged 0 to 17. Further, high rates of respiratory symptoms such as wheeze, allergies, bronchitis symptoms and persistent dry cough were reported among both asthmatic and non-asthmatic children. They suggest the possibility for underdiagnosis of asthma in the population under study. The authors conclude that further studies and preventive measure should be implemented. There are a few comments below to improve the paper:
1. In the results section on line 125, the authors stated that there was a statistically significant difference between asthmatic and non-asthmatics for parental education and in Table 1 this was shown to be p=0.08. However the authors state that a statistically significance was defined as p<0.05.
2. The authors invited four schools within 20 miles of the Salton Sea to participate. How many schools were there in the area and was this a representative sample? Can the authors comment on this further?
3. The study was conducted over a period of one year from May 2017-May 2018 in four schools over a two week period. Are there seasonal changes in this region and can the authors comment further as to data collection for the individual schools were collected in different seasons?
4. In the Discussion, line 193. The authors state that the overall prevalence of asthma found in the population study (22,4%) was higher than other studies in children aged 0 to 17. This is a large age range, and the mean age of your population study was 7.6 (SD ± 0.86). Are there any published studies for a closer age group to this study?
5. In the Discussion, line 248. The author state that maternal smoking during pregnancy increased the odds of child asthma, but there was small number of mothers who reported smoking did not lead to reach the statistical significance. While the OR results are shown in Table 3, we do not know the actual number of mothers who reported smoking in this population. Can this information be included?
6. In the discussion on line 192, the authors state that there is planned implementation of a water transfer agreement that will dramatically accelerate the shrinking of the saline lake. This is the first time this information is mentioned and as it would possibly have significant impacts on health it may be relevant to include this in the introduction. This would support repeating the study after the water transfer agreement has been implemented.
7. In the Results, Figure 1, shows the prevalence of respiratory symptoms among the asthmatic and non-asthmatic participants. However, in the description on line 149 the authors state that the 'Bars represent 149 percentage of parents/guardians of asthmatics (n=80; black bars) and non-asthmatics (n=277; gray 150 bars) who reported having any of the listed respiratory symptoms in the past 12 months'. This indicates that the symptoms are related to the parents/guardians which is confusing.
8. Figure 2 on page 7 is missing a legend or description.
9. Table 3 should be Table 2 as there is no Table 2.
Author Response
Reviewer 2
Please see our responses below each point in italics.
In the results section on line 125, the authors stated that there was a statistically significant difference between asthmatic and non-asthmatics for parental education and in Table 1 this was shown to be p=0.08. However the authors state that a statistically significance was defined as p<0.05.
We thank the reviewer for pointing out this error. We have now removed parental education from this description of statistically significant results.
The authors invited four schools within 20 miles of the Salton Sea to participate. How many schools were there in the area and was this a representative sample? Can the authors comment on this further?
Yes, the school selection was representative of this area. We chose one school from each town within approximately 20 miles of sea. In three of the four towns represented, the selected school was the only elementary school present in the town. The fourth town has 4 elementary schools and one of these schools was selected as a representative school with input from our community partners. We have now included further details about school selection in the methods.
The study was conducted over a period of one year from May 2017-May 2018 in four schools over a two week period. Are there seasonal changes in this region and can the authors comment further as to data collection for the individual schools were collected in different seasons?
We have now clarified in the Methods that the periods during which surveys were administered were May 2017 and May 2018, and all questions referred to symptoms within the past 12 months. Therefore we don't have information to test the impact of seasonal influences, but plan to explore this factor in future work.
In the Discussion, line 193. The authors state that the overall prevalence of asthma found in the population study (22,4%) was higher than other studies in children aged 0 to 17. This is a large age range, and the mean age of your population study was 7.6 (SD ± 0.86). Are there any published studies for a closer age group to this study?
We have now included additional recent information from the CDC in this section to indicate that our prevalence rates are much higher than national averages for children of a similar age range (5-11 years). Additionally, in paragraph 2 of the Discussion, we attempted to describe our study in the context of other studies with similar demographic profiles, given that asthma rates have been shown to vary widely across Hispanic populations. Similarly high rates of asthma have been observed in a study of 4th and 5th graders in El Paso, but much lower rates have been observed in another study of 5th grade children in the Nogales border area. We have now added details to this paragraph to indicate studies of comparable age ranges for comparison.
In the Discussion, line 248. The author state that maternal smoking during pregnancy increased the odds of child asthma, but there was small number of mothers who reported smoking did not lead to reach the statistical significance. While the OR results are shown in Table 3, we do not know the actual number of mothers who reported smoking in this population. Can this information be included?
The number of children with a biological mother who smoked during pregnancy was indeed small (n=16, 4.5%). These numbers are included in Table 2 of this manuscript.
In the discussion on line 192, the authors state that there is planned implementation of a water transfer agreement that will dramatically accelerate the shrinking of the saline lake. This is the first time this information is mentioned and as it would possibly have significant impacts on health it may be relevant to include this in the introduction. This would support repeating the study after the water transfer agreement has been implemented.
We thank the reviewer for this important point and have now included a line in the discussion to indicate that this work is indeed ongoing and follow-up of the children may help elucidate the role of environmental factors in respiratory health in this area. While this paper was designed to focus on the demographic and lifestyle predictors of asthma, future work will incorporate environmental factors that may impact respiratory health.
In the Results, Figure 1, shows the prevalence of respiratory symptoms among the asthmatic and non-asthmatic participants. However, in the description on line 149 the authors state that the 'Bars represent 149 percentage of parents/guardians of asthmatics (n=80; black bars) and non-asthmatics (n=277; gray 150 bars) who reported having any of the listed respiratory symptoms in the past 12 months'. This indicates that the symptoms are related to the parents/guardians which is confusing.
We have now modified the language of the figure legend for clarity.
Figure 2 on page 7 is missing a legend or description.
It appears that there was an issue in the display of this figure legend in the reformatted manuscript version provided to reviewers and we apologize for this error. The current version of the manuscript includes a legend for Figure 2.
Table 3 should be Table 2 as there is no Table 2.
It appears that there was an issue in the display of Table 2 in the reformatted manuscript version provided to reviewers and we apologize for this error. The original version of this manuscript included three primary tables, as well as three supplemental tables. We have included all tables in this revised version for re-review.